# Mind–Body Exercise for Anxiety and Depression in COPD Patients: A Systematic Review and Meta-Analysis

**DOI:** 10.3390/ijerph17010022

**Published:** 2019-12-18

**Authors:** Zaimin Li, Shijie Liu, Lin Wang, Lee Smith

**Affiliations:** 1School of Wushu, Chengdu Sport University, Chengdu 610041, China; lizaimin_1112@163.com; 2School of Physical Education and Sport Training, Shanghai University of Sport, Shanghai 200438, China; 1911111017@sus.edu.cn; 3Department of Physical Education, Wuhan University of Technology, Wuhan 430070, China; 4The Cambridge Centre for Sport and Exercise Sciences, Anglia Ruskin University, Cambridge CB1 1PT, UK; lee.smith@anglia.ac.uk

**Keywords:** mind–body exercise, COPD, anxiety, depression

## Abstract

Objectives: Mind–body exercise has been generally recognized as a beneficial strategy to improve mental health in those with Chronic Obstructive Pulmonary Disease (COPD). However, to date, no attempt has been made to collate this literature. The aim of the present study was to systematically analyze the effects of mind–body exercise for COPD patients with anxiety and depression and provide scientific evidence-based exercise prescription. Methods: both Chinese and English databases (PubMed, the Cochrane Library, EMBASE, Web of Science, Google Scholar, Chinese National Knowledge Infrastructure, Wanfang, Baidu Scholar) were used as sources of data to search randomized controlled trials (RCT) relating to mind–body exercise in COPD patients with anxiety and depression that were published between January 1982 to June 2019. 13 eligible RCT studies were finally used for meta-analysis. Results: Mind–body exercise (tai chi, health qigong, yoga) had significant benefits on COPD patients with anxiety (SMD = −0.76, 95% CI −0.91 to −0.60, *p* = 0.04, I^2^ = 47.4%) and depression (SMD = −0.86, 95% CI −1.14 to −0.58, *p* = 0.000, I^2^ = 71.4%). Sub-group analysis indicated that, for anxiety, 30–60 min exercise session for 24 weeks of health qigong or yoga had a significant effect on patients with COPD who are more than 70 years and have more than a 10-year disease course. For depression, 2–3 times a week, 30–60 min each time of health qigong had a significant effect on patients with COPD patients who are more than 70 years old and have less than a 10-year disease course. Conclusions: Mind–body exercise could reduce levels of anxiety and depression in those with COPD. More robust RCT are required on this topic.

## 1. Introduction

The Global Initiative for Chronic Obstructive Lung Disease (GOLD) defines chronic obstructive pulmonary disease (COPD) as a disease state characterized by exposure to noxious agents resulting in airflow limitation that is not fully reversible, causing shortness of breath and significant systemic effects [1,2,3], mainly involving the lung, but also can cause systemic (or extrapulmonary) adverse reactions [2], with high disease rate, high disability rate, high mortality rate, and a long course of disease [3]. Smoking, dust, chemicals inhaled, air pollution, and respiratory tract infection have been identified as key determinants for the development of COPD [1]. Moreover, COPD is associated with several comorbidities, such as cardiovascular disease, skeletal muscle dysfunction, osteoporosis, anxiety, and depression [2]. The presence of anxiety and depression has been found to increase the number of acute exacerbations and hospitalization, and reduce the quality of life of COPD patients [4]. Anxiety and depression of COPD are a series of psychological changes of patients, including spirits down, lacking interests, asthenia, anorexia, sleep disorders, and so on, which would lead to suicidal thoughts or behaviors [5]. Relevant data showed that the morbidity rate of depression in the stable stage of a COPD patient was 10%–42%, and that, in acute exacerbation, was 10%–86%. The morbidity rate of anxiety in non-hospitalized settings was 13%–46%, and that in hospitalized settings was 10%–55% [6]. According to the global burden of disease report, in 2015, the prevalence of COPD was 45% higher than 1995, with more than 170 million people diagnosed with the condition in 2015 [7], ranking COPD which will be ranked in the top five of the world’s economic burden of disease [8], and the third leading cause of death globally [9].

In view of the serious morbidity and mortality of COPD and the economic burden, there is a growing body of research on utilizing exercise to alleviate the physiological and psychological pain of patients and improve the quality of life. Li et al. used walking exercise to intervene with COPD patients, compared the experimental group and the control group. It was found that exercise training can effectively relieve the dyspnea, improve anxiety and depression symptoms, and improve the quality of life of COPD patients [10]. De-Silva et al. conducted a randomized controlled study on COPD patients with upper extremity resistance exercise training and found that upper extremity resistance exercise training can significantly improve the exercise ability, respiratory muscle strength, and quality of life [11]. Mind–body exercise, as a way to promote physical and psychological health, has received recent attention in the scientific literature. Mind–body exercise focuses on mind, body, psychology, and behavior, including breathing and physical exercise, meditation, and so on [12,13]. It is characterized by gentle and slow exercise (low-to-medium intensity aerobic exercise), coordination of body and breathing, which is represented by Chinese traditional sports tai chi, health qigong, and Indian yoga [14,15,16,17]. Compared with other forms of sports, mind–body exercise (tai chi, health qigong, yoga) is easy to learn and practice, which has not high requirements for equipments and spots [18,19,20,21]. Emphasizing trinity of mind, body, and breathing [22,23], it has the advantages of physical and psychological exercise [24,25,26,27,28]. An increasing number of studies have showed that mind–body exercise has significant effects on improving rheumatoid arthritis [29], cardiovascular disease [30], primary osteoporosis [31], musculoskeletal pain and sleep disorders [32,33], anxiety, depression, and mental health [24,34,35].

In addition, a large number of studies have been carried out on the influence of mind–body exercise on COPD with anxiety and depression [36,37,38,39,40,41,42,43,44,45,46,47,48]. Because of the differences of intervention event (tai chi, health qigong, yoga), samples, time, frequency, duration, the specific effects on COPD patients with anxiety and depression and other negative psychology is not clear. Therefore, the present work aims to carry out a systematic review to evaluate the effects of mind–body exercise on COPD patients with anxiety and depression.

## 2. Method 

### 2.1. Search Strategy

Literatures are searched using the following databases: PubMed, the Cochrane Library, EMBASE, Web of Science and Google Scholar, Chinese National Knowledge Infrastructure, Wanfang, and Baidu Scholar. Databases were searched using the following terms: Tai Chi, Taiji, Tai Chi Chuan, Health qigong, Qigong, Qi Gong, Chi Kung, traditional Chinese exercise, yoga, meditative movement, chronic obstructive pulmonary disease, COPD, chronic obstructive lung disease, chronic obstructive airway disease, emphysema, chronic airflow limitation, chronic airway obstruction, depression, anxiety, mental health, mental problem, psychological health, and psychological problem. Only studies published between January 1982 to June 2019 were considered.

### 2.2. Inclusion and Exclusion Criteria

Inclusion criteria: (1) randomized controlled trail(RCT); (2) peer-reviewed articles published from January 1982 to June 2019; (3) tai chi, health qigong, yoga was the main intervention; (4) studies include at least one measurement of anxiety and depression outcome; (5) study participants are from 45 to 85 years; (6) study participants must meet the inclusion criteria for patients in the Global Initiative for Chronic Pulmonary Diseases; (7) the control group was non-exercise or the exercise was not tai chi, health qigong, yoga.

Exclusion criteria: (1) duplicated studies; (2) reviews, observational studies, abstract only articles, and non-RCT studies; (3) no data or not clear reported for analysis.

According to the inclusion and exclusion criteria, two reviewers independently screened out studies by reading titles and abstracts, removed duplicates, and finally confirmed the eligible studies for this systematic review. If there is divergence between two reviewers, a third reviewer would be the participant to solve the problem.

### 2.3. Study Selection and Data Extraction 

Two reviewers independently conducted literature screening, data extraction, and cross-checking. The extraction information includes: first author, published year, language, participant characteristics, frequency of the intervention, time of each intervention session, duration of the intervention, outcome measure, and follow-up. A third reviewer (W.L.) resolved disagreements between the primary two reviewers.

### 2.4. Quality Assessment of Each Eligible Study

The Physical Therapy Evidence Database (PEDro) scale was used to assess the quality of each eligible study [15], including 11 items as follows: evaluation of eligibility criteria, randomization, concealed allocation, similar baseline, blinding of participants, blinding of instructors, blinding of assessors, more than 85% retention, missing data management (intention-to-treat analysis), between-group comparison, point measure, and measures of variability. Blinding of instructors and participants is impossible during this type of exercise intervention. Therefore, for the present study, this item was removed from the scale. Finally, in all RCT, patients had to be diagnosed with COPD to be included. Therefore, this evaluation of eligibility criteria was moved. Furthermore, mind–body exercise as a primary intervention is possibly combined with other treatments like usual care, which could affect the explanation of study findings. Thus, isolated exercise intervention was added in our quality assessment. Therefore, 9 items were used to assess the quality of each eligible study. Higher scores mean higher methodological quality of study.

### 2.5. Study Analysis Method

Stata 14.0 (Manufacturer, City, US State abbrev. if applicable, Country) was used to carry out meta-analysis: effect combination, heterogeneity analysis, regression analysis, sensitivity analysis, and mapping overall forest plot. For continuous variables, the standardized mean differences (SMD) and 95% confidence interval (95% CI) were used. I^2^ < 50% was deemed as low heterogeneity, and a fixed effect model was adopted. I^2^ ≥ 50% was deemed as a high heterogeneity, and a random effect model was adopted. In the present study, regression analysis will be used whether a study has high or low heterogeneity, if the high heterogeneity existed, the purpose was to find out the heterogeneous source; if the heterogeneity was low, the purpose was to find out the main factors affecting the results. Sub-group analysis was also used in this study to find out which subgroup is more effective for COPD patients with anxiety and depression.

## 3. Results

### 3.1. Studies Selection

A total of 253 articles were found from 8 databases and other resources. After duplicates were removed, 249 articles remained. A total of 223 articles were removed after title and abstracts screen for non-relevant articles (*n* = 218), and abstract-only (*n* = 5). The remaining 26 articles were further screened by reading full-text article. 13 studies were removed owing to non-randomized controlled trials (*n* = 1), review (*n* = 5), and no or unclear outcome measures (*n* = 7). 13 eligible studies were included for our meta analysis (Figure 1). 

### 3.2. Characteristics of Eligible Studies

There were 13 eligible RCT studies in English and Chinese, which trialed in China [37,41,42,43,44,45,46,47,48], America [36,38], Australia [39], and India [40]. There were 906 participants in total, 451 participants were in the mind–body exercise group, and 455 participants were in the control group, the largest sample was 132 participants [37], and the smallest sample was 10 participants [36]. The age of participants ranged from 36 to 83 years, and the intervention duration range from 8 to 48 weeks, 2–7 times a week, 30–90 min each time. Among 13 eligible studies, there were 3 tai chi studies [36,39,47], 3 yoga studies [38,40,45], and 7 health qigong studies in 13 eligible articles [37,41,42,43,44,46,48]. Among this, 3 studies used follow-up assessment to determine if the long-term beneficial effect of mind–body exercise on anxiety and depression [37,39,47] (Table 1).

### 3.3. Methodological Quality Assessment

Scores for the methodological quality of all eligible studies ranged from 4 to 9, as shown in Table 2. All studies were RCT, and carried out similar baseline, between-group comparison, point measure, and measures of variability description. Only 3 studies had concealed allocation and blinding of assessors implementation [37,39,40], 1 study had less than 85% retention [38], and 2 studies were non-isolate exercise intervention [42,48]. Only 2 studies reported how missing data were managed [36,39].

### 3.4. Meta-Analysis of Outcome Indicators

#### 3.4.1. Effect of Mind–Body Exercise on COPD Patients with Anxiety

11 articles compared the difference of anxiety between the exercise group and the control group before and after the experiment among the 13 included articles. Figure 2 shows that the meta-analysis of 11 studies demonstrates mind–body exercise to have significant effects on reducing anxiety in COPD patients (SMD = −0.76, 95% CI −0.91 to −0.60, *p* = 0.04, I^2^ = 47.4%). For depression outcome indicator, all 13 studies were included. Figure 3 demonstrates that the meta-analysis of 13 studies showed mind–body exercise to have significant effects on reducing depression in COPD patients (SMD = −0.86, 95% CI −1.14 to −0.58, *p* = 0.000, I^2^ = 71.4%).

#### 3.4.2. Regression Analysis

Covariates (disease course, age, frequency, time, duration and event) are likely to be the influencing factors for COPD patients with anxiety and depression. The results of regression of covariates for COPD patients with anxiety and depression are presented in Table 3 and Table 4. For anxiety, there were no significant effects for disease course (95% CI −0.0806571 to 0.0526403, *p* = 0.591), age (95% CI −0.0176077 to 0.0821531, *p* = 0.147), frequency (95% CI −0.2993696 to 0.01210669, *p* = 0.304), time (95% CI −0.0088187 to 0.033514, *p* = 0.181), duration (95% CI −0.0340436 to 0.0834159, *p* = 0.308), event (95% CI −0.4031051 to 0.6963495, *p* = 0.5). For depression, there were also no significant effects for disease course (95% CI −0.174951 to 0.238143, *p* = 0.721), age (95% CI −0.0372465 to 0.0928345, *p* = 0.336), frequency (95% CI −0.6229385 to 0.2576581, *p* = 0.349), time (95% CI −0.0296165 to 0.0471513, *p* = 0.596), duration (95% CI −0707587 to 0.0350514, *p* = 0.441), event (95% CI −0.873095 to 0.959349, *p* = 0.912).

#### 3.4.3. Sub-Group Analysis

In all eligible articles, we divided the objects into different groups according to event, disease course, age, frequency, time, duration, as shown in Table 5 and Table 6. Results from sub-group analysis are as follows: (1) for COPD patients with anxiety, both health qigong and yoga had significant effects. For COPD patients with depression, health qigong was significant. (2) disease course: for COPD patients with anxiety, mind–body exercise was more effective for the patients of more than 10 years anxiety. However, for COPD patients with depression, by contrast, mind–body exercise was of greater benefit to the patients of less than 10 years with the condition. (3) age: mind–body exercise revealed it is beneficial for both COPD patients greater than 70 years with anxiety and depression. (4) frequency: mind–body exercise showed a great contradiction for COPD patients with anxiety, the effect of 2–3 times a week, 6–7 times were better than 4–5 times a week, and for COPD patients with depression, the effect of 2–3 times a week were more effective. (5) time: 30–60 min per exercise session significantly improved the symptom of anxiety and depression in COPD patients. (6) duration: 24-week intervention duration showed a significant effect on COPD patients with anxiety, but for depression, intervention duration was not important.

## 4. Discussion

This present systematic review and meta−analysis aimed to collate and analyze the literature pertaining to RCT on mind−body exercise in relation to its influence on COPD patients with anxiety and depression, in order to find out the optimal exercise prescription. The results suggest that mind−body exercise (tai chi, health qigong, yoga) can reduce both anxiety and depression in COPD patients. Under the condition of no adverse reaction, medical workers could combine mind–body exercise with usual medical care to optimize the treatment of anxiety and depression in COPD patients.

The findings in this systematic review are in accordance with the conclusions of previously−published systematic reviews investigating the effects of mind–body exercise (e.g., yoga, health qigong) on alleviating COPD patients with anxiety and depression [49,50]. Specifically, the effect size of anxiety was −0.91 to −0.60 (*p* < 0.01), and depression was −1.14 to −0.58 (*p* < 0.01). Meanwhile, a significant heterogeneity may have been present in the Yang et al. study [45]. They demonstrated a large effect on depression in COPD when compared with effect size of other single studies. Through the analysis of the study, it was revealed that the intervention (self−created yoga breathing exercise) and disease course (T = 17.19 ± 8.20, C = 15.50 ± 5.89) of COPD patients are different from other studies. From the event intervention aspect, the yoga they used was different from traditional yoga [45]. From the disease course aspect, we found that the average disease course of participants was greater than 15 years. Interestingly, one study suggested that the longer disease course is, the less effective the treatment of depression will be [51], and our sub-group analysis of depression also showed the same point. In general, more RCTs are required to reinforce the assumption that self-created yoga tailored to the participant could improve depression in COPD patients with the condition of greater than 15 years.

Sub-group analysis indicated that, for anxiety, 30–60 min per session for 24 weeks of qigong or yoga had a greater effect on patients with COPD patients who were more than 70 years or older with more than 10-year disease course. In the subgroup analysis of frequency, both 2–3 times a week and 6–7 times a week were found to be significant. For depression, 2–3 times a week, 30−60 min each time of qigong had a higher effect on patients with COPD who were greater than 70 years and have less than 10-year disease course. This finding is consistent with results of previous meta analysis, which revealed that at least 3 times a week for more than 30 min each time for 25−36 weeks of qigong is most effective for depression [52]. It is worth mentioning that, in sub-group analysis of anxiety and depression, tai chi did not show a strong influence compared with yoga and qigong, while yoga showed differences in heterogeneity in both anxiety sub-groups (I^2^ = 0.0%) and depression sub-groups (I^2^ = 89.1%). The small number of studies with tai chi and yoga as the main event might be the main reason for the deviation of results.

Studies have showed that anxiety and depression in COPD patients may be partly caused by systemic inflammatory response [53], hypoxia [54] and neuronecrosis of hippocampus [47]. (1) Excessive smoking triggers COPD patients’ pulmonary injury, which stimulates the release of Tumor Necrosis Factor-α(TNF-α), Nitric Oxide(NO), Interleukin-8(IL-8) and other inflammatory mediators [55], which can directly regulate the metabolism of 5-hydroxytryptamine, noradrenaline, dopamine, and other neurotransmitters in the central nervous system, consequently affecting emotional and conscious motor dominated areas of the brain and inducing anxiety and depression [56]. Moreover, anxiety increases sympathetic nervous tension and releases catecholamine [53]. (2) Chronic hypoxia, CO_2_ retention, and acidosis can damage brain cells and cerebrovascular, causeleukoencephalopathy and striatal frontal lobe dysfunction, which promotes anxiety and depression disorder in COPD patients [57]. (3) Glucocorticoid receptors are abundant in the hippocampus. Redundant glucocorticoid content can lead to degeneration and necrosis of hippocampal neurons, and induce depression [50]. Importantly, a series of factors, such as physical activity limitation, social activity impairment, physical deterioration, and social and economic factors’ interaction, likely result in emotional disorders of COPD patients [58]. Mind–body exercise emphasizes the combination of physical and breathing activities, the exerciser can reduce the breathing frequency by relying on his own breathing muscle strength, which can increase the body’s intake of oxygen [59], thus improving the patient’s hypoxia condition. In the process of exercise, it can also increase tryptophan hydroxylase, supply 5-hydroxytryptamine synthesis [60], activate the endogenous cannabinoid system, change the function of hypothalamus pituitary adrenal axis, increase the level of noradrenaline [61], decline glucocorticoid receptor [62], and regenerate the hippocampal neurons [63,64]. Taichi has been shown to inhibit the activity of the left hemisphere, promote the excitation of the right hemisphere, and consequently increase the pleasant mood [65]. Another important factor is that mind–body exercise can improve a series of physiological symptoms of COPD patients, such as exercise capacity and dyspnea [21,66,67,68,69]. The improvement of physical health indicators also plays an important role in guiding the improvement of anxiety and depression symptoms. Therefore, the decrease of anxiety and depression level observed in this systematic review may be attributed to physical and mental exercise (tai chi, health qigong, yoga), which not only emphasizes the coordination of body and breath, but also emphasizes the realization through body and breath and mind in the process of practice [70].

In the present review, two articles showed that there was no significant difference effect on COPD patients with anxiety and depression. Specifically, Gloria Y et al. showed that, compared to the control group, there was no significant trends toward depression [36]. Donesky D et al. indicated that there were no significant changes within or differences between the yoga and the usual group in depressive symptoms and state of anxiety [38]. Both of the studies have the following characteristics: (1) small sample size. Gloria Y et al. just included only 10 participants. Donesky D et al. had 41 subjects in the study, but the withdrawal rate was 7.1%, with the problem of small sample size also existing. (2) The intervention time, frequency, and duration of two studies were twice a week, 60 min each time, lasted for 12 weeks. Exercise “dose” is not strong enough, which is also the reason why it cannot bring significant clinical changes in the measurement results.

## 5. Limitations

Although there were some articles of the present systematic review from Asian and European countries, a majority of studies were carried out in China. As the origin of tai chi, qiong, and yoga, Asian researchers are superior to European researchers in event characteristics, practice methods, and essentials, which would be the reason why European study of mind–body was less than Asian. Certainly, mind–body exercise, as an effective event, should be the research object of the world. So, the cooperation between Asian and European is necessary. Besides, the eligible articles of our study show a variation in quality: (1) only 3 studies displayed their concealed allocation, which would lead to systematic bias of therapeutic effectiveness [71]. (2) Only 3 researches referred to blinding of assessors. (3) The deficiency of the description of disease course for some studies led to failure of recognizing effectiveness of mind–body in COPD patients with anxiety and depression. That is to say, complete disease course information could help understand the effectiveness of mind–body exercise in different COPD stages, and we could give COPD patients better exercise prescription. (4) During the intervention, most of the studies just used mind–body exercise as their rehabilitation means rather than mind–body exercise with medical care, which has a negative impact on patients’ better recovery. And, the difference of experimental design, time, frequency, duration, and outcome measure method would lead to difference outcomes, cause explanation difficulty. At the same time, some studies treat anxiety and depression as the second outcome to analysis, which whether mind–body exercise suitable for COPD patients with anxiety and depression remains to be verified.

## 6. Conclusions

This study used meta-analysis to further strengthen the positive effects of mind–body exercise on anxiety and depression in COPD patients. However, the shortcomings of the research included in the experimental design also have an impact on the results. Therefore, more scientific and reasonable randomized controlled trials should be designed to prove the effectiveness of mind–body exercise on anxiety and depression in COPD patients.

## Figures and Tables

**Figure 1 ijerph-17-00022-f001:**
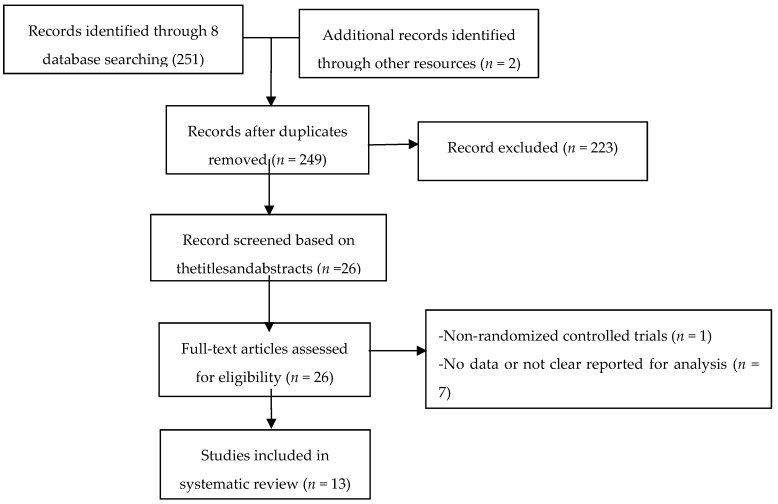
Flow of study selection.

**Figure 2 ijerph-17-00022-f002:**
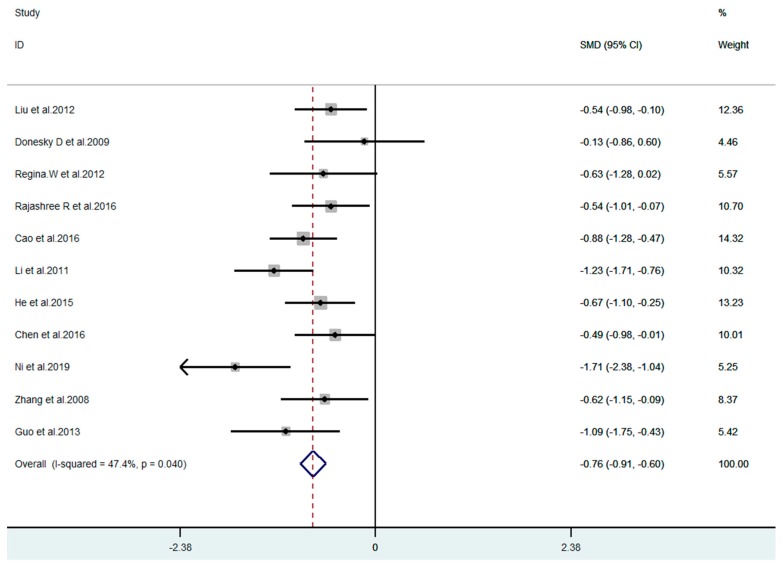
Effect of mind–body exercise on anxiety.

**Figure 3 ijerph-17-00022-f003:**
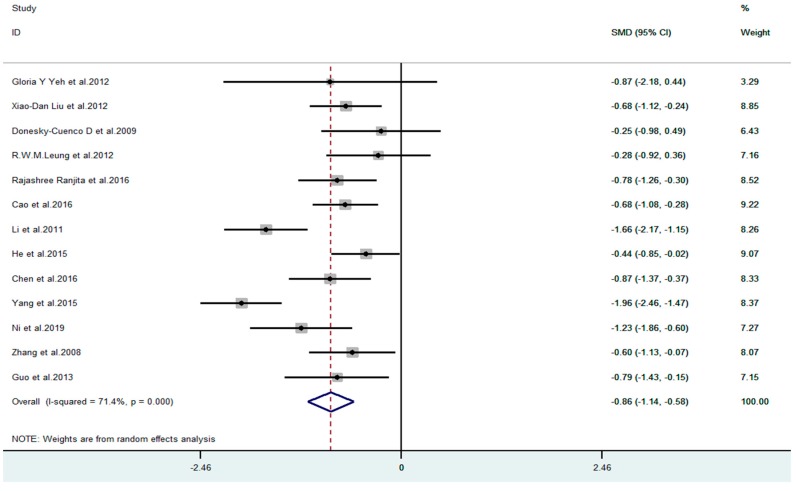
Effect of mind–body exercise on depression.

**Table 1 ijerph-17-00022-t001:** Summary characteristics of included studies.

Reference	Location (Language)	Participant Characteristics			Intervention Program	Training			Outcome Measured	Follow-Up
		Sample Size (AttritionRate)	Disease Course	Mean Ageor Age Range		Frequency (Weekly)	Time (Min)	Duration (Week)		
Gloria Y et al. [36]	Boston, USA	10	T = 2.4 ± 0.5	T = 65 ± 6	T = tai chi + usual care	2	60	12	CESD	no
	(English)	0%	C = 2.6 ± 0.5	C = 66 ± 6	C = usual care					no
Liu et al. [37]	Nanjing, China	132	T = 7.54 (2.73)	T = 61.82 (7.69)	T = yijinjing, wuqinxi, liuzijue, baduanjin,	3	60	24	QoL	yes
	(English)	8.90%	C = 7.57 (2.97)	C = 62.2 (6.34)	C = health education					yes
Doranne D et al. [38]	San Francisco, USA	41	NR	T = 72.2 ±6.5	T = yoga + usual care control	2	60	12	CESD	no
	(English)	7.10%	NR	C = 67.7 ±11.5	C = usual care control + education pamphlet				SSAI	no
Regina W et al. [39]	Sydney Australia	42	T = 8	T = 75(83−67)	T = sun-style tai chi	5	30	12	HADS	yes
	(English)	9%	C = 8	C = 75(83−67)	C = usual medical care					yes
Rajashree R et al. [40]	Karnataka, India	81	T = 9.92 ± 3.25	T = 53.69 ± 5.66	T = yoga	6	90	12	BDI STAI	no
	(English)	8.90%	C = 10.69 ± 2.54	C = 54.36 ± 5.40	C = conventional therapy					no
Cao et al. [46]	Nanjing, China	103	T = 13.33 ± 9.39	T = 70.83 ± 6.22	T = usual care + Baduanjin	4	30	24	SAS SDS	no
	(Chinese)	9.90%	C = 14.54 ± 7.61	C = 70.14 ± 5.71	C = usual + walking					no
Li et al. [47]	Chengdu, China	80	T = 10.56 ± 5.7	T = 64.56 ± 4.73	T = usual care + taijiquan	5	30	12	SAS SDS	no
	(Chinese)	0%	C = 9.80 ± 6.12	C = 62.68 ± 5.76	C = usual care					no
He et al. [41]	Bozhou, China	100	T = 14.52 ± 5.96	T = 58.66 ± 7.56	T = usual care + wuqinxi	4	30-60	24	QoL	no
	(Chinese)	9.30%	NR	C = 58.64 ± 7.52	C = usual care					no
Chen et al. [42]	Qidong, China	70	NR	T = 68.75 ± 8.67	T = usual care + rehabilitation method + liuzijue	7	60	24	HADS	no
	(Chinese)	9.57%	NR	C = 69.31 ± 7.54	C = usual care + rehabilitation method					no
Yang et al. [45]	Wuhan, China	98	T = 17.19 ± 8.20	T = 63.70 ± 5.69	T = usual care + yoga breathing exercise	7	48	48	SDS	yes
	(Chinese)	9.18%	C = 15.50 ± 5.89	C = 64.49 ± 6.10	C = usual care					yes
Ni et al. [43]	Shenyang, China	50	T = 8.09 ± 3.23	T = 51.78 ± 4.02	T = usual care + wuqinxi	5	60	12	HAMA HAMD	no
	(Chinese)	9.40%	C = 7.92 ± 3.41	C = 51.08 ± 4.49	C = usual care					no
Zhang et al. [48]	Shanghai, China	57	NR	T = 69.9 ± 4.73	T = usual care + relaxing exercise + baduanjin	3	30	8	SAS SDS	no
	(Chinese)	0%	NR	C = 69.68 ± 8.66	C = usual care					no
Guo et al. [44]	Nanjing, China	42	T = 6.52 ± 2.43	T = 56.68 ± 6.42	T = daily life + health qigong	NR	NR	24	QoL	no
	(Chinese)	0%	C = 6.82 ± 2.43	C = 58.94 ± 5.96	C = daily life					no

Note: T = trait group, C = control group, NR = not reported, CES − D = The Center for Epidemiological Studies Depression Scale, QoL = Quality of life score, SSAI = The Center for Epidemiological Studies Depression Scale, HADS = The Hospital Anxiety and Depression Scale, BDI = Beck Depression Inventory, STAI = State and Trait Anxiety Inventory, SAS = self-rating anxiety scale, SDS = self-rating depression scale, HAMA = Hamilton anxiety scale, HAMD = Hamilton depression scale.

**Table 2 ijerph-17-00022-t002:** Study quality assessment for eligible randomized controlled trails.

Author [Reference]	Item 1	Item 2	Item 3	Item 4	Item 5	Item 6	Item 7	Item 8	Item 9	Score
Gloria Y et al. [36]	1	0	1	0	1	1	1	1	1	7
Liu et al. [37]	1	1	1	1	1	0	1	1	1	8
Doranne D et al. [38]	1	0	1	0	0	0	1	1	1	5
Regina W et al. [39]	1	1	1	1	1	1	1	1	1	9
Rajashree R et al. [40]	1	1	1	1	1	0	1	1	1	8
He et al. [41]	1	0	1	0	1	0	1	1	1	6
Chen et al. [42]	1	0	1	0	1	0	1	1	0	5
Ni et al. [43]	1	0	1	0	1	0	1	1	1	6
Guo et al. [44]	1	0	1	0	1	0	1	1	1	6
Yang et al. [45]	1	0	1	0	1	0	1	1	1	6
Cao et al. [46]	1	0	1	0	1	0	1	1	1	6
Li et al. [47]	1	0	1	0	1	0	1	1	1	6
Zhang et al. [48]	1	0	1	0	1	0	1	1	0	4

Note: Item 1 = randomization; Item 2 = concealed allocation; Item 3 = similar baseline; Item 4 = blinding of assessors; Item 5 = more than 85% retention; Item 6 = missing data management(intention-to-treat analysis); Item 7 = between-group comparison; Item 8 = point measure and measures of variability; Item 9 = isolate exercise intervention; 1 = explicitly described and present in details; 0 = absent, inadequately described, or unclear.

**Table 3 ijerph-17-00022-t003:** Regression analysis of covariate for COPD patients with anxiety.

_ES	Coef.	Std.err.	t	P > t	(95% Conf. Interval)	
age	0.046262	0.020389	2.27	0.064	−0.0036268	0.0961507
disease course	0.028787	0.027162	1.06	0.33	−0.0376763	0.095201
frequency	−0.18836	0.080357	−2.34	0.058	−0.3849857	0.0082662
time	0.009947	0.007121	1.4	0.212	−0.0074775	0.0273719
duration	−0.019	0.012111	−1.57	0.168	−0.0486287	0.0106379
event	0.159324	0.144056	1.11	0.311	−0.1931674	0.5118153
_cons	−3.66007	1.646477	−2.22	0.068	−7.688854	0.3687162

**Table 4 ijerph-17-00022-t004:** Regressionanalysis of covariate for COPD patients with depression.

_ES	Coef.	Std.err.	t	P > t	(95% Conf. Interval)	
age	0.031596	0.084411	0.37	0.721	−0.174951	0.238143
disease course	0.027794	0.026581	1.05	0.336	−0.0372465	0.0928345
frequency	−0.18264	0.17994	−1.02	0.349	−0.6229385	0.2576581
time	0.008767	0.015687	0.56	0.596	−0.029616	0.0471513
duration	−0.01785	0.021621	−0.83	0.441	−0.0707587	0.0350514
event	0.043127	0.37444	0.12	0.912	−0.873095	0.959349
_cons	−2.24179	2.063563	−1.09	0.319	−7.291147	−2.807567

**Table 5 ijerph-17-00022-t005:** Sub-group analysis of COPD patients with anxiety.

Group	Sub-Group	N	SMD	95% CI	*p*	I^2^
event	tai chi	2	−1.02	−1.41, −0.64	0.145	53%
	health qigong	7	−0.77	−0.96, −0.59	0.076	47.50%
	yoga	2	−0.42	−0.82, −0.03	0.356	0.0%
disease course (year)	less than 10 years	5	−0.79	−1.03, −0.54	0.033	61.9%
	more than 10 years	3	−0.90	−1.15, −0.65	0.224	33.2%
year	50–59.9	4	−0.86	−1.12, −0.59	0.028	67%
	60–69.9	4	−0.72	−0.96, −0.48	0.11	50.3%
	more than 70	3	−0.68	−1.00, −0.37	0.216	34.80%
frequency (time/week)	2–3 times/week	3	−0.5	−0.8, −0.19	0.548	0.0%
	4–5 times/week	5	−0.96	−1.18, −0.74	0.063	55.1%
	6–7 times/week	2	−0.52	−0.86, −0.18	0.889	0.0%
time (min)	30 ≤ min < 60	5	−0.83	−1.04, −0.61	0.372	6.1%
	60 ≤ min ≤ 90	5	−0.63	−0.87, −0.4	0.016	67.3%
duration (week)	8–12 weeks	6	−0.82	−1.06, −0.59	0.01	67%
	24 weeks	5	−0.71	−0.91, −0.5	0.511	0.0%

**Table 6 ijerph-17-00022-t006:** Sub-group analysis of COPD patients with anxiety.

Group	Sub-Group	N	SMD	95% CI	*p*	I^2^
event	tai chi	3	−0.96	−1.97,0.05	0.004	81.9%
	health qigong	7	−0.71	−0.89, −0.52	0.55	0.0%
	yoga	3	−1.02	−2.00, −0.04	0.000	89.1%
disease course (year)	less than 10 years	6	−0.75	−0.99, −0.51	0.483	0.0%
	more than 10 years	4	−1.17	−1.89, −0.46	0.000	90%
year	50–59.9	3	−0.77	−1.19, −0.34	0.111	54.4%
	60–69.9	7	−1.08	−1.52, −0.64	0.000	75.9%
	more than 70	3	−0.51	−0.82, −0.21	0.424	0.0%
frequency (time/week)	2–3 times/week	4	−0.59	−0.89, −0.29	0.758	0.0%
	4–5 times/week	5	−0.85	−1.34, −0.37	0.001	78.4%
	6–7 times/week	3	−1.2	−1.95, −0.46	0.001	85.3%
time (min)	30 ≤ min < 60	6	−0.78	−1.02, −0.55	0.496	0.0%
	60 ≤ min ≤ 90	6	−0.94	−1.48, −0.40	0.000	86.4%
duration (week)	8–12 weeks	7	−0.83	−1.24, −0.42	0.007	66.0%
	24 weeks	5	−0.90	−1.32, −0.47	0.000	79.4%
	48weeks	1	−1.96	−2.46, −1.47	−−	−−

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
