# Peer review of "Mind–Body Exercise for Anxiety and Depression in COPD Patients: A Systematic Review and Meta-Analysis"

_ijerph, 2019, doi:10.3390/ijerph17010022_

Round 1

Reviewer 1 Report

It is really outstanding the fact that mild exercise could help COPD  patients with depression and anxiety. This was well pointed in this paper througt the analysis of 13 studies from internet database. More studies must be held to confirm that depresion and anxiety wich characterizes copd patients and is described in many studies can be reduced by mild daily excersise.

Author Response

Dear reviewer: thanks for your comments and suggestions. Just as you said, more studies must be held to confirm that depression and anxiety which characterizes COPD patients and is described in many studies can be reduced by mild daily exercise. So, in the later studies, we will focus on more similar researches instead of mind-body exercise only, such as the effects of other exercise event for COPD patients with anxiety and depression, or the differences between medical treatment and exercise event in COPD patients with anxiety and depression.

Reviewer 2 Report

This is a very interesting systematic review and qualitative synthesis of mind-body exercise for anxiety and depression in COPD patients. The study may be of potential interest from the clinical standpoint. However the way the article is written need to be revised in order to make it readable.

Minor comments

-The authors should specify the definition of anxiety and depression used in the studies included and the magnitude of the difference consider significant.

-If authors have registered their review, we recommend providing the registration number at the end of the abstract.

-References 8 and 30 do not support what is claimed in the text.

-The inclusion criteria number 7 is the same as number 6.

-Characteristics of eligible studies: when referencing the 7 health qigong studies included in the review, the authors include reference 36 (Effect of yoga training in patients with chronic obstructive pulmonary disease: a systematic review and meta-analysis).

-Table 2. Please line up 0 in column 5.  

- This study should be revised for spelling and grammatical errors (e.g. tears instead of years, line 34,  page 3, analyst instead of analyze).

-At the beginning of page 3 , when describing results from sub-group analysis , point (1) is missing.

Author Response

Dear reviewer: thanks for your comments and suggestions.

The authors should specify the definition of anxiety and depression used in the studies included: Anxiety and depression of COPD are a series of psychological changes of patients, including down spirits, interests-lacking, asthenia, anorexia, sleep disorders and so on, which would lead to suicidal thoughts or behaviors. We did not register reviews. I’ve changed reference 8 and 30. The inclusion criteria number 7 has been removed. Reference 36 was change into correct reference. I’ve lined up 0 in column 5, table 2. Spelling and grammatical errors were revised. At the beginning of page 3, when describing results from sub-group analysis, the missing point (1) was added.

Reviewer 3 Report

Dear authors,

As a person that practices regularly sport, I consider that exercises bring substantial benefits for both the body and the mind thus it was a great pleasure reading your manuscript.

After carefully reading your manuscript I consider that it is suitable for publication in the actual form but it would be great if you could include something from the following questions:

I see that the majority of your references are from Asia and it is clearly that you have a different culture thus the bias. Why do you think that Europeans have such a reluctance towards this body-mind approach? How could other European countries implement and improve this approach?

It would be interesting for the reader to find these questions and possible answers in your manuscript.

Author Response

Dear reviewer: thanks for your comments and suggestions. Based on your question, I’ve some ideas as follows: first, I’ve no culture bias when I choose the references. The reason why my references were Asian researchers is that taichi, qigong, and yoga were originated from Asia. Asian researchers are superior to European researchers in event characteristics, practice methods and essentials, which would be the reason why European study of mind-body was less than Asian. Certainly, mind-body exercise, as an effective event, should be the research object of the world. So, the cooperation between Asian and European is necessary.